# Relation between Adaptive Eating and Energy Intake Coping Strategies in a Refeed Model for Bodybuilders

Wilson Max Almeida Monteiro de Moraes [1,2,*], Ronaldo Ferreira Moura [1], Ragami Alves [3], José de Oliveira Vilar Neto [4], Bruno Magalhães de Castro [1], Douglas Leão [1] and Jonato Prestes [1]

[1] Graduation Program on Physical Education, Catholic University of Brasilia, Brasilia 71966900, Brazil; ronaldomoura.personal@gmail.com (R.F.M.); brunodemagalhaes@gmail.com (B.M.d.C.); doug.leaop@gmail.com (D.L.); jonatop@gmail.com (J.P.)

[2] Graduation Program on Nutrition, Center University of Maua (UniMauá), Brasilia 71966900, Brazil

[3] Post Graduation Program in Physical Education, Federal University of Parana, Curitiba 81531-980, Brazil; ragami1@hotmail.com

[4] Physical Education and Sports Institute, Federal University of Ceara, Fortaleza 60020181, Brazil; jvilarr@gmail.com

[*] Correspondence: wmaxnutri@gmail.com; Tel.: +55-61-9919-46507

**Abstract:** Lean bodybuilder athletes may encounter challenges in adapting their eating habits during ad libitum refeed, either intuitively or consciously. Aims: This paper investigates whether there is a relationship between adaptive eating and energy intake coping strategies in a refeed model for bodybuilders. Methods: Fourteen male bodybuilders ($29.6 \pm 3.1$ years; $85.6 \pm 6.8$ kg, $\geq 6$ competitions) completed a 4-week regimen consisting of 5 days of energy restriction followed by 2 days of refeed. Dietary assessment, body composition (ultrasound), recovery stress questionnaire (REST-Q) and Brunel mood scale (BRUMS) were utilized pre- and post-regimen. Coping function questionnaire (CFQ), mindful eating scale version 2 (MES 2) and the intuitive eating scale-2 (IES-2) were evaluated at the 4th week. Results: Compared to the initial values, the refeed day resulted in a daily caloric increase of 44% compared to the average energy intake on the energy restriction days, culminating in a weekly calorie deficit of 27% and a drop in body mass of $3.1 \pm 1.4\%$. Most participants showed reduced body fat and preserved or gained lean mass. The energy consumption during the refeed maintained an inverse relationship with the perception of satiety ($r = -0.9$; $p < 0.01$), the IES 2 total scores ($r = -0.82$; $p < 0.01$), as well as the confidence in hunger and satiety cues ($r = -0.62$; $p = 0.02$) and congruence in food–body choice dimensions ($r = -0.56$; $p = 0.04$). Emotional coping maintained an inverse relationship with the IES 2 total scores ($r = 0.54$; $p < 0.05$) and an inverse relationship with energy intake during refeed ($r = -0.42$; $p < 0.05$). Conclusion: The results suggest that a heightened perception of internal hunger and satiety signals and higher scores in intuitive eating may contribute to adequate energy intake, even when high scores of emotional coping are present.

**Keywords:** physique athletes; refeed; intermittent energy restriction; intuitive eating; coping

## 1. Introduction

Bodybuilding (BB) is a modality in which competitors are judged in terms of muscular appearance, symmetry, and leanness in proportional physiques during rounds of poses in a contest [1]. In the pre-competitive period (PreC), usually 8 to 26 weeks before the competition, bodybuilders commonly combine energy restriction (ER) and increased energy expenditure to adjust body mass to a target weight class, reducing body fat stores as much as possible, while maintaining or modestly gaining fat-free mass (FFM) [1,2].

Energy restriction can be conducted either continuously or intermittently (IER). Continuous energy restriction requires reducing energy intake each day below what is needed for weight maintenance, whereas IER alternates periods of restriction with periods of higher energy intake in a non-linear fashion [3]. One of the most popular configurations of IER

among strength athletes is energy restriction for 5–6 consecutive days following one or two days following high-carbohydrate and -energy intake (similar to maintenance levels or slightly higher (~5 to 10% above requirements)) [4].

Adherents of this approach suggest that it may be advantageous because of the greater availability of carbohydrates and energy, improving mood, motivation and performance [5]. In agreement with the possible physiological and psychological benefits of IER, Peos et al. [6] demonstrated that interruption of energy restriction for a week increased lower limb muscle resistance and reduced the subjective perception of hunger in resistance-trained individuals. Furthermore, it has been shown that the addition of cheat meals, defined as a planned consumption of a favorite food that is not part of the prescribed or regular training diet [4], could provide a better affective response and attenuation in objective and subjective markers of muscle recovery after a protocol based on high-volume training in bodybuilders [4].

Although bodybuilders are recognized in the literature for having rigid attitudes following food selection, meal frequency and supplementation [7], it is not clear whether refeed practices can optimize the proposed energy intake. Some individuals admit that refeed practices can be a good opportunity to consume "forbidden foods", reporting overfeeding and energy intake consistent with a compulsive episode [8]. Supposedly, the dichotomy "forbidden or not" of restraint suggests a behavior consistent with rigid and inflexible diets, which could present difficulty for adaptive eating (intuitively or consciously) or disconnecting internal physiological signals of hunger and satiety to the detriment of emotional reasons [9]. Supporting this, higher scores of intuitive eating correlate with lower disordered eating behaviors and disinhibition episodes [10].

The PreC is particularly critical for athletes undergoing dietary restraint. In this line, we demonstrated previously that bodybuilders undertaking ~44% energy restriction presented a higher perception of stress and worse mood states in comparison with a period of ~15% positive energy balance [2], while Hickey et al. [11] reported that higher hunger was associated with stress indicators and poor athletic performance in student athletes.

The ability to deal with stressors, referred to as coping, in theory leads to food behaviors as a form of self-regulation to stress, and, thus, the athlete can develop more adaptive psychophysiological responses [10,12]. Interestingly, individuals tend to eat in response to distress and dysphoric emotions, particularly when coping is focused in emotions (hereafter emotional cooping), which are more passive, not promoting adaptive and problem-changing behaviors [12].

In view of the above, it is not clear whether athletes undergoing energy restriction could benefit from refeed practices in a pre-competitive context and whether energy intake arising from refeed is related to coping strategies. The purpose of the present study was to test the following hypotheses: (1) whether refeed practices may contribute to adequate energy intake in bodybuilders with more adaptive eating (conscious and intuitive eating pattern) and (2) whether higher levels of coping focused on emotion are associated with maladaptive eating and, possibly, a higher energy intake.

## 2. Methods

### 2.1. Participants

The present study selected participants through convenience sampling methodology. The president of a local federation optimized the contacts for participation in the study. Fourteen male bodybuilders completed data collection, which occurred during the last week in the off season and during four initial weeks following PreC period.

After receiving an explanation of the study, the athletes who agreed to participate were evaluated regarding their habitual nutritional intake, and then a personalized diet was prescribed based on an energy deficit in order to reach the weights corresponding to their categories [4]. The inclusion criteria were aged 18–40 years; participation in at least three contests and in preparation for one competition. Sleep data (hours total and habits) were collected; and social jetlag was calculated as the absolute difference between mid-sleep

on weekends and mid-sleep on weekdays [13]. Exclusion criteria included participants reporting diuretics or the utilization of laxatives, appetite inhibitors as well as those who did not present regularity in training during the collection of data. The research was approved by the Ethics Committee of Catholic University of Brasilia (process 3664095) in accordance with the Helsinki Declaration, and all participants signed an informed consent form.

### 2.2. Anthropometric Data

All anthropometric measurements were performed before and after the 4-week follow-up. A balance Plenna® and Stadiometer (Alturexata, São Paulo, Brazil) were utilized for body weight and height measurements, respectively.

Body composition was determined by a 2.5 MHz A-mode transducer portable ultrasound (BodyMetrix, BX2000, IntelaMetrix, Inc., Livermore, CA, USA) to determine the sum of the subcutaneous adipose tissue thickness values at seven standardized sites: triceps, subscapular, chest, axillary, suprailiac, abdominal and thigh [14,15]. All measurements were performed by the same evaluator with experience in handling the device [16], and the average of three measurements was used for analysis.

### 2.3. Energy Intake

Each athlete completed at least one 3-day food diary each week to assess current nutritional intake according to the household measurement. Nutritional data were processed using the Webdiet® software (https://pt.webdiet.com.br/). Data were adjusted for body mass and expressed in grams and percent calories of total energy. Energy deficit was estimated as 40% lower than the habitual intake observed at the end of the off-season period during restricted days.

Food preferences were used to develop individual meal plans. The quality and quantity of food sources were controlled, with care taken to maintain adequate proportions between the main macronutrients. During each week of the 4-week study period, participants consumed the allocated energy restriction for five days followed by two days of refeed. On refeed days, participants were instructed to have 2 daily meals ad libitum and record their food intake, as previously described [4].

The adaptive eating questionnaires were obtained (Figure 1. Design of the study) in the fourth week.

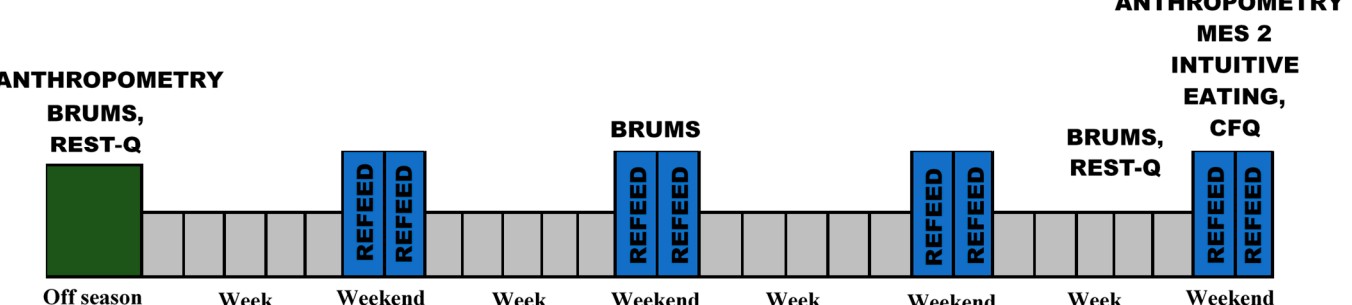

**Figure 1.** Design of the study. BRUMS: Brunel mood scale; REST-Q; recovery stress questionnaire for athletes; CFQ: Coping Function Questionnaire; MES 2: Mindful Eating Scale version 2.

### 2.4. Psychological Distress and Mood Disturbance

Restrictive-period distress was assessed through the recovery stress questionnaire for athletes (RESTQ-Sport), translated and validated to Brazilian Portuguese [17]. Measurements of simultaneous frequency of the actual stress with the frequency of recovery-associated activities were performed before and after the 4-week follow-up. RESTQ-Sport includes 76 items distributed in 19 scales (10 related to stress and 9 related to recovery). Each scale contains four items evaluated by a 6-point Likert-type question (ranging

from 0 = "never" to 6 = "always"). Final scores were calculated as the sum of the stress-related scales ($\Sigma S$) and recovery ($\Sigma R$), and the difference between $\Sigma R$ and $\Sigma S$.

Mood states were measured using the Brunel mood scale (BRUMS) questionnaire, previously translated into Portuguese and validated in a Brazilian population [18]. The BRUMS is a self-report questionnaire consisting of 24 items rated on a 5-point scale (ranging from 0 = "no" to 4 = "extremely") designed to assess 6 dimensions, each consisting of 4 items. The total score of each dimension ranges from 0 to 16. The results were expressed as total mood disturbance, which was determined by the sum score of the negative dimension subtracting the score of the positive dimension vigor and then adding the value "100". The BRUMS was utilized before and after the 4-week follow-up.

### 2.5. Adaptive Eating
2.5.1. Mindful Eating

The Mindful Eating Scale version 2 (MES 2) questionnaire [19] was used to assess mindfulness, especially in the domain of eating behavior. The MES 2 contains 28 items with response options on a Likert-type scale, with 1 being considered "never" up to 4 "always".

The MES 2 consists of five domains: (1) consciousness (consciousness of what the food looks like, and consciousness of how it tastes); (2) distraction (when attention is not focused on food); (3) disinhibition (continues to eat even when satisfied); (4) emotional response (eating in response to sadness/stress); (5) external influences (in response to external cues, such as advertisements). The higher the MES 2 score, the greater the ability to eat mindfully.

2.5.2. Perception of Hunger and Appetite

To assess the perception of hunger/appetite, individuals utilized a hunger odometer [20]. This instrument is a scale with numerical values from 0 to 10, with lower values corresponding to greater perception of hunger and less satiety; values between 4 and 6 correspond to an interval that represents a comfortable moment to have a meal in order to respect the body, not letting it starve and not exceeding the feeling of satiety; and above 6, there is a greater perception of satiety to the detriment of hunger. A reminder for access was sent via the Web diet application®.

2.5.3. Intuitive Eating

For intuitive eating, individuals answered the Intuitive Eating Scale-2 (IES-2), translated and validated into Brazilian Portuguese [21]. The scale consists of 23 items with the following dimensions: unconditional permission to eat (UPE), eating for physical rather than emotional reasons (EPR), confidence in hunger and satiety signals (RHSC) and body-food congruence (B-FCC). High scores indicate better levels of intuitive eating.

### 2.6. Coping Construct

To measure the coping construct, the Coping Function Questionnaire (CFQ) [22] for Brazilian athletes was used. The score was determined by the average of the answers given to the items of each dimension (problem-focused coping (6 items), emotion-focused coping (7 items) and avoidance-focused coping (5 items). Items are rated on a 5-point Likert scale ranging from 1 = not at all to 5 = very much.

### 2.7. Statistical Analysis

The parameters were described with mean and standard error of the mean. To test the normality of data distribution, the Shapiro–Wilk test was performed. When necessary, paired *t*-tests were used for comparison between timepoints; Spearman's correlation was utilized to assess the association between variables. Significance was considered when $p < 0.05$.

## 3. Results

Young male athletes were selected to participate in this study. The mean age was $29.9 \pm 1.2$ years with $10.5 \pm 1.1$ years of training experience; all competed in bodybuilding events ($6.6 \pm 0.2$ contests).

Table 1 presents food intake, anthropometric parameters and training and sleep characteristics of the subjects.

**Table 1.** Mean daily macronutrient and energy intake during 4 weeks of energy restriction and subject characteristics regarding anthropometry, training and sleep. Values are expressed as mean and standard deviation. * paired test t, $p < 0.05$.

| Food Intake | Weeks | | | |
|---|---|---|---|---|
| | **1** | **2** | **3** | **4** |
| Protein (g/kg) | $2.8 \pm 0.5$ | $2.6 \pm 0.3$ | $2.7 \pm 0.5$ | $2.6 \pm 0.4$ |
| % energy | $38.0 \pm 4$ | $40 \pm 4$ | $36 \pm 4$ | $37 \pm 4$ |
| Carbohydrate (g) | $236 \pm 36$ | $242 \pm 29$ | $231 \pm 32$ | $249 \pm 36$ |
| % energy | $35 \pm 4$ | $35 \pm 3$ | $35 \pm 3$ | $35 \pm 4$ |
| Fats (g) | $83 \pm 6$ | $78 \pm 6$ | $88 \pm 7$ | $88 \pm 6$ |
| % energy | $27 \pm 4$ | $25 \pm 4$ | $29 \pm 5$ | $28 \pm 3$ |
| Energy intake (kcal) | $2729 \pm 132$ | $2801 \pm 148$ | $2746 \pm 129$ | $2731 \pm 124$ |

| Anthropometric parameter | Pre | Post |
|---|---|---|
| Height (cm) | $173.0 \pm 0.1$ | $173.1 \pm 0.1$ |
| Body mass (kg) | $85.6 \pm 6.8$ | $83.5 \pm 5.9$ * |
| Body fat (%) | $7.3 \pm 0.4$ | $4.0 \pm 0.2$ * |
| Fat mass (kg) | $6.1 \pm 0.3$ | $5.0 \pm 0.2$ * |
| Lean body mass (kg) | $79.4 \pm 5.9$ | $78.4 \pm 5.6$ * |

| Training caractheristics | | |
|---|---|---|
| Resistance training | | |
| Days/week | $5.9 \pm 0.3$ | $6.3 \pm 0.2$ * |
| Minutes/week | $517.5 \pm 23.2$ | $635.5 \pm 29.2$ * |
| Poses training | | |
| Days/week | $4.6 \pm 0.8$ | $5.0 \pm 0.6$ * |
| Minutes/week | $79.3 \pm 10.7$ | $89.3 \pm 12.5$ * |
| Aerobic exercise | | |
| Days/week | $4.5 \pm 0.6$ | $5.6 \pm 0.2$ * |
| Minutes/week | $261.4 \pm 35.4$ | $327.4 \pm 30.2$ * |

| Sleep | | |
|---|---|---|
| Hours/day | $6.4 \pm 1.6$ | $5.9 \pm 1.4$ |
| Social jet leg | $1.1 \pm 0.7$ | $1.2 \pm 0.8$ |

Weekly energy restriction over four weeks was ~27%. The energy intake during refeed days was ~44% higher than energy intake over restricted days. It was estimated for refeed days that ~70% of the energy total was provided by carbohydrates. Protein intake corresponded to a mean of 2.6 g/kg (minimum 2.2 and maximum 3.1 g/kg) per day. In general, the frequency of meals (6–7/day) was the same in restricted and refeed days with two athletes consuming one less meal during refeed days.

With the exception of one athlete, all individuals reduced body fat, and the majority preserved or gained lean mass ($n = 11$). Weight loss was $2.5 \pm 1.4$ kg, corresponding to 3.1% in relation to initial values and weekly weight loss of ~0.8%. For those athletes who lost weight at a rate greater than 0.5% per week, there was a correlation between the number of calories ingested during refeed and the weekly weight loss rate ($r = 0.7$; $p < 0.05$).

Although it was not the main objective of the study, it is interesting to note that energy intake was associated with social jetlag but not total sleep time. Furthermore, individuals

who had social jetlag > 1 (n = 8) reduced less body fat over the 4 weeks when compared to individuals with social jetlag ≤ 1. In general, the athletes ate 5–7 meals per day.

Mood states and stress recovery are presented in Figure 2. In the fourth week of energy restriction, bodybuilders displayed poor mood in relation to the final off-season period, as observed for increased levels in scores for total mood in BRUMS. Additionally, dimensions from REST-Q (Figure 2B) general stress and sport stress were increased during energy restriction in comparison to the final off-season period. Both the general and sport recoveries were lower in energy restriction in comparison to the off-season period.

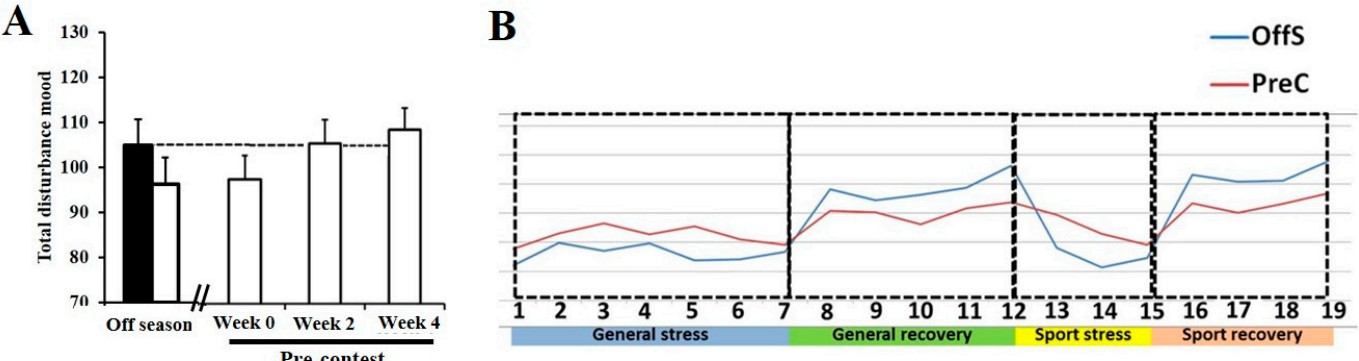

**Figure 2.** Bodybuilders' and control students' total disturbance mood and REST-Q survey results during off-season and pre-contest time points. Panels (**A,B**): REST-Q sports questionnaire values are expressed as mean and standard error. Black bars signify an age-matched group of non-athletic students. White bars given for athletes.

Table 2 presents data regarding adaptive eating, expressed in total scores and dimensions respective for mindful eating, perception of hunger/appetite and intuitive eating.

**Table 2.** Total scores and dimensions of the Mindful Eating Scale 2 (MES), hunger and appetite scale and intuitive eating (n = 14). Values are expressed as mean and standard deviation.

| MES 2 Scale | |
|---|---|
| Consciousness | 1.8 ± 0.2 |
| Distraction | 2.3 ± 0.2 |
| Disinhibition | 2.1 ± 0.1 |
| Emotional response | 2.0 ± 0.1 |
| External influences | 1.5 ± 0.1 |
| Total score | 9.5 ± 0.7 |
| Perception of hunger/appetite | 4.8 ± 0.4 |
| IES 2 scale | |
| Unconditional permission to eat | 14.9 ± 0.6 |
| Eating for physical rather than emotional reasons | 24.0 ± 0.6 |
| Confidence in hunger and satiety signals | 17.5 ± 1.1 |
| Body-food congruence | 9.2 ± 0.7 |
| Total scores | 65.9 ± 1.4 |

Eight athletes (57%), when starting the cheat meal, described scores between 4 and 6 on the hunger/satiety scale, while three (21%) perceived "more hunger", and three (21%) reported being predominantly satiated.

There was no correlation between total score and the dimensions in the MES 2 and energy intake ($p > 0.05$).

As seen in Figure 3, energy consumption during the cheat meal maintained an inverse relationship with the perception of hunger ($r = -0.9$; $p < 0.01$), with total scores of intuitive

eating (r = −0.82; *p* = 0 < 0.01), as well as with the RHSC (r = −0.62; *p* = 0.02) and BFCC (r = −0.56; *p* = 0.04) dimensions.

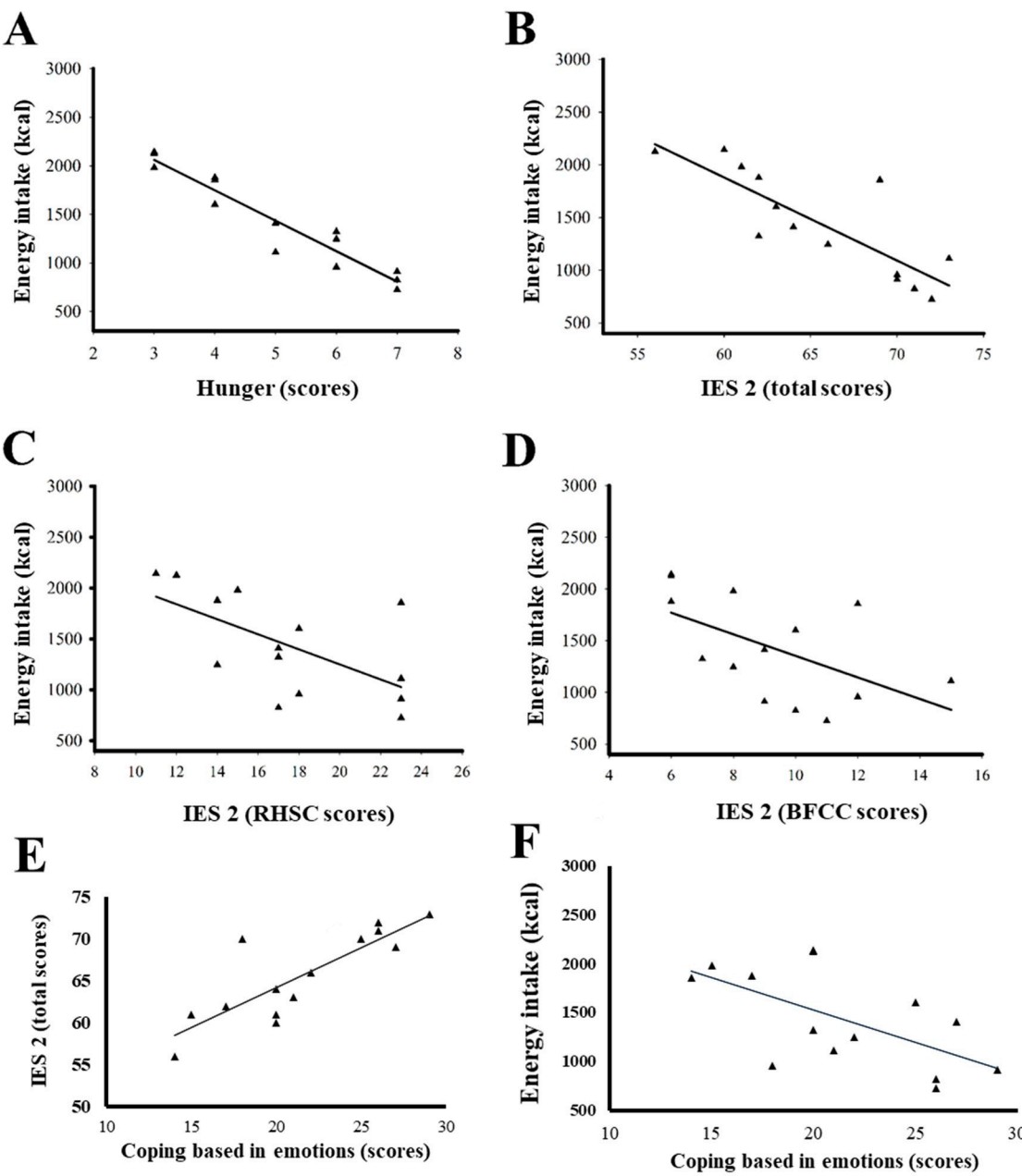

**Figure 3.** Linear relations between bodybuilders' energy intake and adaptive eating. Panel (**A**): Energy intake during cheat meal and hunger/satiety perception scores. Panel (**B**): Energy intake during cheat meal and intuitive eating scores (IES 2). Panel (**C**): Energy intake during cheat meal and scores of the dimension confidence in hunger and satiety cues (RHSC) dimension. Panel (**D**): Energy intake during cheat meal and scores of the congruence in food–body choice (BFCC) dimension. Panel (**E**): IES 2 scores and emotional meal coping. Panel (**F**): Energy intake during cheat meal.

In addition, coping based on emotions maintained an inverse relationship with the total scores of IES 2 (r = 0.54; *p* < 0.05) and energy intake during the cheat meal (r = 0.40; *p* < 0.05).

## 4. Discussion

Based on the premise that athletes could consume more energy than the proposed amount during refeed, this study examined the relation between adaptive eating and coping based on emotions with energy intake in a model of refeed for bodybuilders. The main findings were as follows: (1) most athletes consumed energy adequately; (2) higher scores of intuitive eating and perception of satiety cues were associated with lower energy intake; and (3) higher scores of coping based on emotions were associated with higher scores of intuitive eating and lower energy intake.

In recent years, bodybuilding athletes have implemented refeed strategies, temporarily reverting to ad libitum intake, increasing caloric intake in order to achieve energy or positive balance in a weight loss plan [3]. Although refeed can be organized in several formats, in the present study, we used a common configuration among athletes (based on their preliminary reports) and similar to other authors [23], which consisted of alternating 5 days of energy restriction and 2 days of refeed (5:2). The refeed day resulted in a daily caloric increment of ~44% compared to the average energy intake on the energy restriction days. Thus, the strategy culminated in a weekly calorie deficit of ~27%, suggesting moderate energy restriction [23].

Of interest, most athletes (10/14 or 71%) adjusted the proposed energy intake, consuming food of their preference. In fact, a study of our group demonstrated that athletes maintained relatively stable energy intake with cheat meals over 4 weeks of energy restriction with refeed on weekends (5:2) [4]. As reported by Syed-Abdul et al. [24], while IER may presumably lead to a reduction in energy intake for most of the week, ad libitum carbohydrate refeed may not be sufficient for compensating energy deficit on restricted days.

Despite refeed having the possibility of contributing adequate energy intake, this strategy may be valid if it effectively contributes to greater flexibility and adherence to the diet plan. We observed no drop out in participants in our study; however, it is interesting to note that the follow-up was only 4 weeks. Peos et al. [25] demonstrated that the dropout rate for the intervention with IER was about two-times lower than the dropout rate observed with 12 weeks of continuous energy restriction. This response can mainly be attributed to the nature of a non-prescribed diet on days of diet break. However, it is also interesting to note that diet breaks, when taken too far, can be harmful, similar to what is consistent with disordered eating behaviors and disinhibition episodes [8]. For bodybuilders, this is of particular interest since energy restriction is necessary for a "cutting" and leaner physique, which determines the performance aesthetic in a contest[1]. Future research should investigate the effects of how IER may interact with restrained eating patterns in bodybuilders.

In the context of sport, restricting the diet and classifying foods as prohibited are common [9]. Thus, intuitive eating practices can be disrupted, leading to a lower awareness of hunger and satiety in athletes, which may lead to cravings for these forbidden foods, especially when deprived. Furthermore, sport is associated with intense emotional experiences, and as such, the potential for emotionally triggered cravings is high, resulting in emotional eating at the expense of physical hunger [9].

Of interest, athletes in the present study had extremely low levels of body fat (4%) in the fourth week, which was less than previously reported in a systematic review [26]. These characteristics are consistent with those found in experienced athletes and may reflect the long participation of bodybuilding practices in the current participants. Particularly interesting was the fact that the majority of athletes reduced body fat, since both the loss of body fat and the gain or preservation of leaner mass are more difficult when lower body fat levels are present [27]. Additionally, an inverse correlation between the amount of calories ingested during refeed and weight loss when the weight loss rate is superior to 0.5% weekly suggests that rapid weight loss makes it difficult for body recomposition. In fact, it has been suggested that slower rates of weight loss ($\leq$0.5% of body mass/week) are preferable for attenuating the loss of fat-free mass in leaner competitors [28].

Of note, athletes who start the refeed with more satiety tend to consume less energy. Thus, the use of the perception of hunger/satiety scale could be an interesting strategy, helping to guide athletes consuming cheat meals in periods that correspond to a lower perception of hunger, especially those with greater difficulty in connecting the internal signs of hunger and satiety.

The negative correlation between energy consumption during refeed and intuitive eating scores confirms our hypothesis and corroborates the findings of Herbert et al. [29], in which it was shown that intuitive eating scores were positively correlated with interceptive sensitivity scores (ability to recognize body processes). Additionally, Plateau, Petrie and Papathomas [30] showed that intuitive eating practices, including three principles permission to eat, recognition of hunger and satiety signals, and eating to satisfy physical and nutritional needs, helped to reduce tendencies towards compulsive episodes in athletes after their competitive career. Taken together, these findings suggest that individuals who eat more intuitively are more aware of what their bodies are "telling them", and using intuitive eating principles may decrease the chance of developing unhealthy eating patterns.

Utilizing a single rigid diet can involve a higher level of stress by limiting food intake, thereby increasing psychological distress or mood disturbances [2,11,31]. Importantly, we observed that both total mood disturbance and perceived stress increase in response to energy restriction during PreC, as observed in BRUMS and REST-Q scores, indicating that decisions including what to eat during the cheat meal are likely to be made under conditions of stress and altered mood.

Coping strategies are used by athletes to deal with stressful events, not only in the contest but during preparation for competitions. We utilized a tool that evaluated coping focused on the problem, avoidance and emotions. Intriguingly, only coping based on emotions showed a relationship with energy intake (Figure 3E). Our findings regarding the relationship between IES 2 and coping based on emotions (Figure 3F) contrast with a report by Deroost and Cserjési [12], in which individuals that utilized coping strategies with more focus on emotions presented more emotional eating, culminating in a more passive strategy. Thus, higher scores in intuitive eating can optimize the adequacy of energy intake from refeed, when high scores of coping based on emotions are present in bodybuilders.

No relationship was observed between the scores related to mindful eating and energy consumption, underscoring the need to recognize that the scale has some limitations, as it has not been previously tested with bodybuilding athletes. Furthermore, the total time of energy restriction (4 weeks) may not have been sufficient to culminate in changes in body weight determinants that could be detected by the MES 2 scale. A direction for future investigation would be to conduct a study in the post-competition period, in which athletes usually exhibit a high frequency of compulsive episodes [32] and display a possible disconnection between physical and emotional hunger. Another limitation of the study was sample size and the lack of control group. However, it is very difficult to carry out studies in high-level bodybuilders, and those who qualify are generally hesitant to change their training practices for the sake of a research study.

Lastly, others factors may influence energy intake during refeed. For example, athletes with higher social jetlag showed a higher energy intake, suggesting that jetlag, in any instance, might have a deleterious effect in the perception of internal hunger and satiety signals. As reported previously by Roenneberg et al. [13], individuals with social jetlag (>1 h) were at greater risk for becoming overweight, even after adjustments for confounding variables (gender, age, sleep duration and chronotype). It is possible that circadian misalignment may play a pivotal role in response to refeed.

The literature has shown that refeeds during energy restriction can benefit the athlete due to the transient increase in lower limb muscle resistance [6,25], better affective response and recovery after a session of a protocol based on a high volume of resistance training [4] and reduced feelings of hunger and irritability, as well as greater satiety [25]. The present investigation demonstrated that extremely lean bodybuilders can optimize the adequacy of energy intake during refeed when high scores of intuitive eating and perception of hunger

and satiety cues are increased. The relationship between coping based on emotions with higher rates of intuitive eating and lower energy intake suggests that emotions and control have a role in energy intake during refeed.

## 5. Conclusions

The higher perception of internal hunger and satiety signals may contribute to adequate energy intake through refeed during energy restriction, suggesting the need for interventions involving the principles of intuitive eating.

Higher levels of coping based on emotion are associated with higher levels of intuitive eating, which are also associated with lower energy intake during refeed. More studies are needed to better understand the relationship between refeed strategies, (mal)adaptive eating and coping strategies.

**Author Contributions:** Conceived and designed the experiments: W.M.A.M.d.M., R.F.M., B.M.d.C. and D.L. Analyzed the data: W.M.A.M.d.M., R.F.M., J.d.O.V.N., R.A. and D.L. Wrote the manuscript: W.M.A.M.d.M., J.d.O.V.N., R.A. and J.P. Critical review manuscript: W.M.A.M.d.M., J.d.O.V.N., R.A., B.M.d.C., D.L. and J.P. All authors have read and agreed to the published version of the manuscript.

**Funding:** This research received no external funding.

**Institutional Review Board Statement:** The study was conducted in accordance with the Declaration of Helsinki, and approved by the Ethics Committee of Catholic University of Brasilia (process 3664095) for studies involving humans.

**Data Availability Statement:** Data are contained within the article.

**Acknowledgments:** The authors are grateful to Larissa Lorrayne S Silva for assistance in data collection.

**Conflicts of Interest:** The authors declare no conflicts of interest.

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
