# Peer review of "Relation between Adaptive Eating and Energy Intake Coping Strategies in a Refeed Model for Bodybuilders"

_2674-0311, doi:10.3390/dietetics3010005_

Round 1

Reviewer 1 Report

Comments and Suggestions for Authors

You have generated an interesting study and report of the associated information. The attached file contains a copy of the manuscript with highlighted areas. Please consider addressing these areas. There are some word choices that need to be modified, some text that are not words with meaning and tense issues. My suggestion is to have the entire document edited for grammar. 

Comments on the Quality of English Language

There are some word choices that need to be modified, some text that are not words with meaning and tense issues. My suggestion is to have the entire document edited for grammar. 

Author Response

We thank the reviewers for the peer review and insightful comments in this area. The revised manuscript for our changes which are included in the updated manuscript as tracked changes and in yellow font colour. 

Reviewer 2 Report

Comments and Suggestions for Authors

This paper examines the relations between energy intake, adaptive eating, coping strategies and refeeds in a small sample of male bodybuilders through surveys and questionnaires. Even though the sample is rather small, the approach and the results are of general good quality, and most certainly of interest, as it would provide essential nutritional strategies to support bodybuilders pre, during their contest period. Nevertheless, the paper needs substantial improvement of the English written text throughout the whole of the manuscript. While the following issues should all be attended to:

Please choose a title, now there seems to be both ‘long’ and ‘short‘ versions.

Please mention the exact number of male bodybuilder participating in your study in the methods section and the Table 1.

‘coping in emotion’ can be replaced with ‘emotional coping’.

Please explain ‘cheat mail’ in introduction and methods.

Do you intend to use the abbreviation BB for bodybuilding throughout the manuscript?

Figure 2 uses Total disturbance mood data collected from control group, of non-athletic students. These control students have not been mentioned in methods, also not as participants. This is not correct, because at least these would need to be age/gender-matched. Please provide data. Additionally, is there a possibility that data for these control student from all the scales over a similar period of time would be available, and could be compared (as a control)?

Please, add an appropriate title to each Figure and Table.

Use ‘social jetlag’

Check punctuation throughout.

Comments on the Quality of English Language

Abstract:

Line 17/18 should read: ‘ Extremely lean athletes might have (intuitively or consciously) difficulty in adaptive eating during ad libitum refeed.’

Line 18/19 should read: ‘This study investigates whether there is a relation between adaptive eating and coping strategies with energy intake in a model of refeed proposed for bodybuilders.

Line 20/21 should read: ‘Fourteen male bodybuilders (29.6±3.1yrs; 85.6±6.8kg, ≥6 competitions) completed a 4-week protocol, consisting out of repeating cycle of a 5 days of energy restriction followed by 2 days of refeed.

Line 22/23 change: ‘Brunel mood scale (BRUMS)’

Line 24 replace ‘was used’ by ‘was applied’

Line 26 replace ‘on’ by ‘during’

Line 27/28: reformulate ‘The most reduced body fat and preserved or gained lean mass.’ , as it is not clear.

Line 28 use ‘inverse relationships’

Lines 33/35 should read: ‘Our results suggest that a higher perception of internal hunger and satiety signals, and higher scores in intuitive eating, may contribute to adequate energy intake (adequately), even when high scores of coping - based in emotions - are present.’ Choose the right position and use of ‘adequate’, the brackets indicate the less correct version.

Introduction:

Line 41 should read: ‘In the pre-competitive’

Line 41 replace ‘previous’ with ‘prior’

Line 43 replace ‘for’ with ‘to’

Lines 44/45 should read ‘reducing stored body fat as much as possible and maintain or moderately gain fat-free mass (FFM).’

Line 51 replace ‘following’ with ‘with’

Lines 53/54 should read ‘IER’s advantages might be caused by the greater availability of carbohydrates and energy improving mood, motivation and performance…’

Please reformulate lines 65/69: ‘Supposedly, the dichotomy “forbidden or not” of restraint revel a behavior consistent with rigid and inflexible diets, in which athletes could have difficulty for adaptive eating (intuitively or consciously), disconnecting the internal physiological signals of hunger and satiety to the detriment of emotional reasons [9].’

Line 71 replace ‘undergoing’ with ‘undergo’

Line 73 replace ‘worsens in’ with ‘worsening of’

Line 75 replace ‘were’ with ‘was’

Line 77 should read ‘in theory leads to food choice intake behaviors’

Line 84 replace ‘refeed’ with ‘refeeds’

Lines 84/88: ‘The purpose of the present study was to test the hypotheses: 1) whether refeed practices may contribute to adequate energy intake in bodybuilders with a more adaptive eating (conscious and intuitive eating pattern) and 2) whether higher levels of coping focused in emotion are associated with a maladaptive eating and, possibly a higher energy intake.’

Methods:

Lines 91/93 suggest to omit: ‘The present study selected participants for convenience due to the specificity needed to verify the proposed conditions. The president of local federation optimized the contacts of the athletes for participated of the study.’

Lines 93/96 should read: ’ Fourteen male bodybuilders previously selected completed the data collection, which occured during the last week of the Off season and during the four initial weeks following the PreC period.’

Line 96 should read ‘participate in the study’

Line 97 should read ‘, then their diet’

Line 99 should read ‘three’

Lines 100/101 should read: ‘contest and be in preparation for one competition. Responses were collected by an experienced questioner, who also collected data referring to sleep (hours total and habits).’

Line 103 should read: ‘Excluded were those athletes using…’

Lines 122/123 reformulate ‘Deficit energy was estimated for reduced 40% of the habitual intake observed in final of the off-season period in restricted days.’

Line 126 replace ‘between’ with ‘of’

Line 130 reformulate ‘One eligible cheat meal at fourth week,’

Lines 136/ should read ‘To characterize the restrictive period regarding distress the Recovery stress questionnaire for athletes (RESTQ-Sport) that was translated and validated to Brazilian Portuguese [17]. It was used to measure simultaneous frequency of the actual stress with the frequency of recovery-associated activities, and was conducted before and after the 4-week follow-up.’

Results:

Lines 191/195 should read: ‘The average age of the  participating young male atheles was 29.9 ± 1.2 years with 10.5 ± 1.1 years of training experience of which all competed in bodybuilding events (6.6 ± 0.2 contests). Table 1 presents their food intake, anthropometric parameters, and training and sleep caractheristics.’

Lines 196/199 should read ‘Table 1. Bodybuilder’s mean daily macronutrient and energy intake during 4 weeks of energy restriction and subject characteristics regarding anthropometry, training and sleep.’

Lines 199/204 should read: ‘The weekly energy restriction during four weeks was ~27%. The energy intake with refeed days was ~44% higher than the energy intake during restricted days. It was estimated For refeed day it was estimated that ~70% of the energy total was provided by carbohydrates. The protein intake corre sponded to a mean of 2.6g/kg per day (minimum 2.2 and maximum 3.1g/kg). In general, the frequency of meals (6-7/ day) was the same comparing restricted and refeed days with two athletes consuming one less meal during refeed day.’

Lines 205/209 should read: ‘With the exception of one athlete, all the individuals reduced their body fat, and most athletes either preserved or gained lean mass (n=11). Weight loss medium was 2.5±1.4kg, corresponding to 3.1% decrease as compared to initial values and a weekly weight loss of ~0.8%. For those athletes who lost weight at a rate greater than 0.5% per week, there was a significant correlation between the amount of calories ingested during refeeds and weekly weight loss rate (r=0.7; p<0.05).’

Lines 201/213 should read: ‘It was interesting to note that the energy intake was associated to social jetlag, but not to total sleep time. Furthermore, individuals who suffered from social jetlag >1 (n=8) reduced less body fat over the 4 weeks when compared to individuals with minor social jetlag ≤1. In general, the athletes eat 5-7 meals per day.’

Lines 214/220 should read: ‘The mood states and stress-recovery are presented in Figure 2. At the 4th week of energy restriction, the bodybuilders showed poor mood in relation to the final of the off season period, as  observed for the increased BRUMS scores for total mood. Additionally, the specific dimensions from REST-Q (figure 2B) like general stress and sport stress were increased during energy restriction in comparison to the final/end of the off season period.’ and rephrase: ‘Both: the recovery general and sport were lower in energy restriction in comparison to the final/end of the off season period.’ Perhaps using ‘end’ instead of ‘final’ would work better here?

Lines 221/222, Figure 2 should have a title: ‘Bodybuilder’s and control students total disturbance mood and REST-Q survey results during off season and pre-contest time points.’

Lines 224/226 should read: ‘Table 2 present the data regarding adaptive eating, expressed in total scores and dimensions respective for mindful eating, perception of hunger/appetite and intuitive eating.’

Line 227/228 should read: ‘Table 2. Bodybuilder’s total scores and dimensions of the Mindful Eating Scale 2 (MES), hunger and appetite scale and Intuitive Eating (n=14). Values are expressed as mean and standard deviation.’

Lines 239/245 should read: Figure 3. Linear relations between bodybuilder’s energy intake and perception scores. Panel A: Energy intake during cheat meal and hunger/satiety perception scores. Panel B: Energy intake during cheat meal and intuitive eating scores (IES 2). Panel C: Energy intake during cheat meal and scores of the dimension confidence in hunger and satiety cues (RHSC) dimension. Panel D: Energy intake during cheat meal and scores of the congruence in food-body choice (BFCC) dimension. Panel E: IES 2 scores and emotional meal coping. Panel F: Energy intake during cheat meal.

Lines 246/247 should read: ‘In addition, emotional coping maintained an inverse relationship with the total IES 2 scores(r=0.54; p<0.05) and energy intake during cheat meal (r=0.40; 247 p<0.05).’

Discussion:

Lines 252/256 should read: ‘Our main findings for the bodybuilders were as follows: 1) most bodybuilder athletes consumed adequate energy intake; 2) the higher scores of intuitive eating and perception of satiety cues in bodybuilders were associated with lower energy intake; and 3) higher scores of emotional coping were associated with higher scores of intuitive eating and lower energy intake.’

Line 261 should read ‘similar to other authors’

Line 266 should read ‘Of interest, most bodybuilder athletes’

Line 273 should read: ‘Despite the fact that refeed may contribute to adequate energy intake, we understand that this strategy can only be valid….

Lines 275/276 should read: ‘In our study we did not observe drop out… only at 4 weeks.

Lines 279/280 should read: ‘However, it is interesting to note that diet-breaks when taken too far, can pathologize, leading to disordered eating disorders…’

Remove the superscript 1 in Line 283.

Lines 291/294 should read: ‘It is interesting to note that the bodybuilding athletes of our study had extremely low basal levels of body fat and at the 4th week meet lower levels (4% of body fat) in comparison to the body builders reported in a recent systematic review (5.8–10.7%) [26]. These characteristics….

Reformulate lines 296/298: ‘Particularly interesting was the fact that major of the athletes reduced body fat since both: loss of body fat and gain or preservation in leaner mass are more difficult when lower body fat levels are present [27].’

Lines 301/302 should read: ‘In fact, it has been suggested that slower rates of weight loss (≤0.5% of body mass/week) are preferable for attenuating the loss of fat-free mass in leaner competitors [28].’

Lines 319/323 should be reformulated to be more clear.

Line 325/326 should read: ‘Coping strategies are comummently used by athletes for to deal with stress events, not only in contest, but during preparation for competition as well.’

Lines 327/328 should read: ‘Intriguingly, only emotional coping based showed arelation with energy intake (Figure 3E).’

Line 328/331 should be reformulated to be more clear. ‘Our findings re garding the relation between IES 2 and coping based in emotions (figure 3F) contrast with the reported by Deroost & Cserjési [12], in which individuals that utilized coping strategies with more with focus in emotions presented a more emotional eat, culminating in a more passive strategy.’

Line 335 should read: ‘The fact that we observed no relation between the scores related to mindful eating and energy consumption….’

If you want other scholars to read section lines 349-369, please pay attention to correct English writing, punctuation, terminology etc. I suggest a major edit for these four paragraphs.

Author Response

We thank the reviewers for the peer review and insightful coments and the revised manuscript for our changes which are included in the updated manuscript as tracked changes and in yellow font colour.

Round 2

Reviewer 2 Report

Comments and Suggestions for Authors

Even though the proofread activities performed are giving a much better experience in reading the last section of the discussion, there are still some minor issues (given below). Major issues: The absence of one adequate manuscript title; and the lack of Figure and Table titles.

Nevertheless, the manuscript improved much, and with some editing, and having MDPI's Nutrients proofread service included, the manuscript is a fine addition to the field.

Lines 1/2: Please choose a title, now there seems to be both ‘long’ and ‘short‘ versions. A good and better long title would be: ‘Relation between adaptive eating and energy intake coping strategies in a refeed model for bodybuilders’

Omit line 4, the short title.

Background: Lean bodybuilder athletes may encounter challenges in adapting their eating habits during ad libitum refeed, either intuitively or consciously. Aims: This paper investigates whether there is a relationship between adaptive eating and energy intake coping strategies in a refeed model for bodybuilders. Methods: Fourteen male bodybuilders (29.6±3.1yrs; 85.6±6.8kg, ≥6 competitions) completed a 4-week regimen consisting of 5 days of energy restriction followed by 2 days of refeed. Dietary assessment, body composition (ultrasound), recovery stress questionnaire (REST-Q) and Brunel mood scale (BRUMS) were assessed pre and post regimen. Coping function questionnaire (CFQ), mindful eating scale version 2 (MES 2) and the intuitive eating scale-2 (IES-2) were evaluated at the 4th week. Results: Compared to the initial values, the refeed day resulted in a daily caloric increase of 44% compared to the average energy intake on the energy restriction days, culminating in a weekly calorie deficit of 27% and a drop in body mass of 3.1±1.4%. Most participants showed reduced body fat and preserved or gained lean mass. The energy consumption during the refeed maintained an inverse relationship with the perception of satiety (r=-0.9; p<0.01), the IES 2 total scores (r=-0.82; p<0.01), as well as the confidence in hunger and satiety cues (r=-0.62; p=0.02) and congruence in food-body choice dimensions (r=-0.56; p=0.04). Emotional coping maintained an inverse relationship with the IES 2 total scores (r=0.54; p<0.05) and an inverse relationship with energy intake during refeed (r=-0.42; p<0.05). Conclusion: The results suggest that a heightened perception of internal hunger and satiety signals and higher scores in intuitive eating may contribute to adequate energy intake, even when high scores of emotional coping are present.

Line 44 should read: ‘as much as possible, and maintain or modestly gain fat-free mass (FFM) [1,2]’

Line 50 replace ‘following’ by ‘of’

Line 58 change to: ’…cheat meals, defined as a planned consumption of a favorite food that is not part of the prescribed or regular training diet, could provide….’ And add references.

Line 80/81 change to: ‘coping is focused in emotions (hereafter emotional cooping), which is more passive’ It is suggested to replace this throughout the manuscript.

Line 85 change to: ‘intake arising from refeed is related to coping strategies.’

Line 86/89 change to: ‘The purpose of the present study was to test the hypotheses: 1) whether refeed practices may contribute to adequate energy intake in bodybuilders with a more adaptive eating (conscious and intuitive eating pattern) and 2) whether higher levels of coping focused in emotion are associated with a maladaptive eating and, possibly a higher energy intake.’

Line 97 change to: ‘…and then a personalized diet was prescribed…’

Lines 122 should read:observed at the end of the off-season period during restricted days.’

Line 130: should read ‘Figure 1. Design of the study.’ or ‘Figure 1. Study design.

Line 195 should read ‘Table 1. Bodybuilders’ mean macronutrient…’ Please note that the week identification in the columns has been wrongly repositioned.

Line 200 replace ‘proved from’ with ‘provided by’

Lines 218/220 should be getting some attention, as it is still not clear and should be rephrased. What do you mean to say here? That both the recovery, general recovery and sports recovery, were lower? It is encouraged to use ‘end’ instead of ‘final’ when referring to the last bit of the off season period.

Lines 222/224 Figure 2 should have a title: ‘Bodybuilder’s and control students total disturbance mood and REST-Q survey results during off season and pre-contest time points.’

Lines 243/249 Figure 3 should have a title, and reduce repetition in the subtext: ‘Figure 3. Linear relations between bodybuilder’s energy intake and perception scores. Panel A: Energy intake during cheat meal and hunger/satiety perception scores. Panel B: Energy intake during cheat meal and intuitive eating scores (IES 2). Panel C: Energy intake during cheat meal and scores of the dimension confidence in hunger and satiety cues (RHSC) dimension. Panel D: Energy intake during cheat meal and scores of the congruence in food-body choice (BFCC) dimension. Panel E: IES 2 scores and emotional meal coping. Panel F: Energy intake during cheat meal.’

Remove the superscript 1 in Line 284.

Comments on the Quality of English Language

Lines 1/2: Please choose a title, now there seems to be both ‘long’ and ‘short‘ versions. A good and better long title would be: ‘Relation between adaptive eating and energy intake coping strategies in a refeed model for bodybuilders’

Omit line 4, the short title.

Background: Lean bodybuilder athletes may encounter challenges in adapting their eating habits during ad libitum refeed, either intuitively or consciously. Aims: This paper investigates whether there is a relationship between adaptive eating and energy intake coping strategies in a refeed model for bodybuilders. Methods: Fourteen male bodybuilders (29.6±3.1yrs; 85.6±6.8kg, ≥6 competitions) completed a 4-week regimen consisting of 5 days of energy restriction followed by 2 days of refeed. Dietary assessment, body composition (ultrasound), recovery stress questionnaire (REST-Q) and Brunel mood scale (BRUMS) were assessed pre and post regimen. Coping function questionnaire (CFQ), mindful eating scale version 2 (MES 2) and the intuitive eating scale-2 (IES-2) were evaluated at the 4th week. Results: Compared to the initial values, the refeed day resulted in a daily caloric increase of 44% compared to the average energy intake on the energy restriction days, culminating in a weekly calorie deficit of 27% and a drop in body mass of 3.1±1.4%. Most participants showed reduced body fat and preserved or gained lean mass. The energy consumption during the refeed maintained an inverse relationship with the perception of satiety (r=-0.9; p<0.01), the IES 2 total scores (r=-0.82; p<0.01), as well as the confidence in hunger and satiety cues (r=-0.62; p=0.02) and congruence in food-body choice dimensions (r=-0.56; p=0.04). Emotional coping maintained an inverse relationship with the IES 2 total scores (r=0.54; p<0.05) and an inverse relationship with energy intake during refeed (r=-0.42; p<0.05). Conclusion: The results suggest that a heightened perception of internal hunger and satiety signals and higher scores in intuitive eating may contribute to adequate energy intake, even when high scores of emotional coping are present.

Line 44 should read: ‘as much as possible, and maintain or modestly gain fat-free mass (FFM) [1,2]’

Line 50 replace ‘following’ by ‘of’

Line 58 change to: ’…cheat meals, defined as a planned consumption of a favorite food that is not part of the prescribed or regular training diet, could provide….’ And add references.

Line 80/81 change to: ‘coping is focused in emotions (hereafter emotional cooping), which is more passive’ It is suggested to replace this throughout the manuscript.

Line 85 change to: ‘intake arising from refeed is related to coping strategies.’

Line 86/89 change to: ‘The purpose of the present study was to test the hypotheses: 1) whether refeed practices may contribute to adequate energy intake in bodybuilders with a more adaptive eating (conscious and intuitive eating pattern) and 2) whether higher levels of coping focused in emotion are associated with a maladaptive eating and, possibly a higher energy intake.’

Line 97 change to: ‘…and then a personalized diet was prescribed…’

Lines 122 should read:observed at the end of the off-season period during restricted days.’

Line 130: should read ‘Figure 1. Design of the study.’ or ‘Figure 1. Study design.

Line 195 should read ‘Table 1. Bodybuilders’ mean macronutrient…’ Please note that the week identification in the columns has been wrongly repositioned.

Line 200 replace ‘proved from’ with ‘provided by’

Lines 218/220 should be getting some attention, as it is still not clear and should be rephrased. What do you mean to say here? That both the recovery, general recovery and sports recovery, were lower? It is encouraged to use ‘end’ instead of ‘final’ when referring to the last bit of the off season period.

Lines 222/224 Figure 2 should have a title: ‘Bodybuilder’s and control students total disturbance mood and REST-Q survey results during off season and pre-contest time points.’

Lines 243/249 Figure 3 should have a title, and reduce repetition in the subtext: ‘Figure 3. Linear relations between bodybuilder’s energy intake and perception scores. Panel A: Energy intake during cheat meal and hunger/satiety perception scores. Panel B: Energy intake during cheat meal and intuitive eating scores (IES 2). Panel C: Energy intake during cheat meal and scores of the dimension confidence in hunger and satiety cues (RHSC) dimension. Panel D: Energy intake during cheat meal and scores of the congruence in food-body choice (BFCC) dimension. Panel E: IES 2 scores and emotional meal coping. Panel F: Energy intake during cheat meal.’

Remove the superscript 1 in Line 284.

Author Response

Even though the proofread activities performed are giving a much better experience in reading the last section of the discussion, there are still some minor issues (given below). Major issues: The absence of one adequate manuscript title; and the lack of Figure and Table titles.

Nevertheless, the manuscript improved much, and with some editing, and having MDPI's Nutrients proofread service included, the manuscript is a fine addition to the field.

Response: We appreciate the comments.

Background: Lean bodybuilder athletes may encounter challenges in adapting their eating habits during ad libitum refeed, either intuitively or consciously. Aims: This paper investigates whether there is a relationship between adaptive eating and energy intake coping strategies in a refeed model for bodybuilders. Methods: Fourteen male bodybuilders (29.6±3.1yrs; 85.6±6.8kg, ≥6 competitions) completed a 4-week regimen consisting of 5 days of energy restriction followed by 2 days of refeed. Dietary assessment, body composition (ultrasound), recovery stress questionnaire (REST-Q) and Brunel mood scale (BRUMS) were assessed pre and post regimen. Coping function questionnaire (CFQ), mindful eating scale version 2 (MES 2) and the intuitive eating scale-2 (IES-2) were evaluated at the 4th week. Results: Compared to the initial values, the refeed day resulted in a daily caloric increase of 44% compared to the average energy intake on the energy restriction days, culminating in a weekly calorie deficit of 27% and a drop in body mass of 3.1±1.4%. Most participants showed reduced body fat and preserved or gained lean mass. The energy consumption during the refeed maintained an inverse relationship with the perception of satiety (r=-0.9; p<0.01), the IES 2 total scores (r=-0.82; p<0.01), as well as the confidence in hunger and satiety cues (r=-0.62; p=0.02) and congruence in food-body choice dimensions (r=-0.56; p=0.04). Emotional coping maintained an inverse relationship with the IES 2 total scores (r=0.54; p<0.05) and an inverse relationship with energy intake during refeed (r=-0.42; p<0.05). Conclusion: The results suggest that a heightened perception of internal hunger and satiety signals and higher scores in intuitive eating may contribute to adequate energy intake, even when high scores of emotional coping are present.

Response: Done.

Lines 1/2: Please choose a title, now there seems to be both ‘long’ and ‘short‘ versions. A good and better long title would be: ‘Relation between adaptive eating and energy intake coping strategies in a refeed model for bodybuilders’

Response: Done.

Omit line 4, the short title.

Response: Done.

Line 44 should read: ‘as much as possible, and maintain or modestly gain fat-free mass (FFM) [1,2]’

Response: Done.

Line 50 replace ‘following’ by ‘of’

Response: Done.

Line 80/81 change to: ‘coping is focused in emotions (hereafter emotional cooping), which is more passive…’ It is suggested to replace this throughout the manuscript.

Response: Done.

Line 85 change to: ‘intake arising from refeed is related to coping strategies.’

Response: Done.

Line 86/89 change to: ‘The purpose of the present study was to test the hypotheses: 1) whether refeed practices may contribute to adequate energy intake in bodybuilders with a more adaptive eating (conscious and intuitive eating pattern) and 2) whether higher levels of coping focused in emotion are associated with a maladaptive eating and, possibly a higher energy intake.’

Response: Done.

Line 97 change to: ‘…and then a personalized diet was prescribed…’

Response: Done.

Lines 122 should read: ‘…observed at the end of the off-season period during restricted days.’

Response: Done.

Line 130: should read ‘Figure 1. Design of the study.’ or ‘Figure 1. Study design.’

Response: Done.

Line 195 should read ‘Table 1. Bodybuilders’ mean macronutrient…’ Please note that the week identification in the columns has been wrongly repositioned.

Response: Done.

Lines 222/224 Figure 2 should have a title: ‘Bodybuilder’s and control students total disturbance mood and REST-Q survey results during off season and pre-contest time points.’

Response: Done.